# Heavy Metals in Soil around a Typical Antimony Mine Area of China: Pollution Characteristics, Land Cover Influence and Source Identification

**DOI:** 10.3390/ijerph20032177

**Published:** 2023-01-25

**Authors:** Xiaoqian Li, Yaning Tang, Xinghua Wang, Xiaodong Song, Jiaxue Yang

**Affiliations:** 1School of Environmental Studies, China University of Geosciences, Wuhan 430074, China; 2State Key Laboratory of Biogeology and Environmental Geology, China University of Geosciences, Wuhan 430074, China; 3Institute of Geological Survey, China University of Geosciences, Wuhan 430074, China

**Keywords:** heavy metal, antimony mine, land use, source identification, risk assessment

## Abstract

To understand contamination characteristics and identify sources of heavy metals in soil affected by complex mine activities, a detailed survey of soil heavy metals from different land cover types was investigated around the Xikuangshan (XKS) antimony mine in south-central China. Soil samples had average concentrations of Sb, As, Cd, Cr, Hg, Pb, Cu, Zn and Ni exceeding their background level in the Hunan province. Sb, As and Cd were the main pollutants. A total of 86.8% of samples were severely polluted, characterized by the Nemerow’s comprehensive index, and 68.4% of samples were of very high potential ecological risk, primarily contributed by Sb, Cd and Hg. Among different land cover patterns, Hg, Pb and Cd concentrations showed a statistically significant difference. The application of Pearson correlation, principal component analysis (PCA) and hierarchical cluster analysis (HCA) combined with spatial interpolation GIS mapping revealed that Ni, Cr and Cu were mainly from natural parent materials, whereas other heavy metals were related to anthropogenic sources. Pb, As and Hg were mainly derived from smelting processes of sulfide minerals in the XKS area. The agricultural practice is the main factor for the accumulation of Cd and Zn, and sphalerite smelting also contributed to high Zn concentrations. Particularly, spatial variation of soil Sb concentrations was affected by multiple factors of complex antimony mine activities related to mining, beneficiation and smelting in the XKS area. These results are useful for the prevention and reduction of heavy metal contamination in soils by various effective measures in typical regions affected by antimony mine activities.

## 1. Introduction

Heavy metals, generally referring to metals and metalloids with a density greater than 4.5 g/cm^3^ such as Cd, Cr, Hg, Pb, Cu, Zn, Ni and As, are considered as the main hazardous trace elements and pollutants that are preferentially monitored and controlled. Soil heavy metal contamination attracts great attention around the world, due to its high toxicity at low content thresholds [1,2], persistence and extreme difficulty in removal by natural degradation [3], and bioavailability [4,5,6,7]. The accumulation of heavy metals in soil can degrade soil quality, disrupt ecological services and create serious risks for human health [8,9,10,11].

The presence and accumulation of heavy metals in soil could be caused by natural and anthropogenic activities. The difference in weathering of geological parent materials is the dominant natural source contributing to the spatial distribution of heavy metals in soil [12]. The main anthropogenic sources are related to industrial, agricultural, traffic and mining activities [12,13,14]. Among these, mining-related activities are considered as one of the most prominent human activities leading to high concentrations of heavy metals in the environment [15,16,17], through mining and smelting wastewater discharge, waste rock and slag heaps, mine tailings and dusts around the mining area without appropriate management.

Identifying potential sources related to mine activities for heavy metal pollution is vitally important for controlling the priority pollutants to regulate soil heavy metal pollution. In recent years, an increasing number of studies have focused on using mathematical models and statistical analysis to identify the potential sources of heavy metals in soils surrounding typical mining areas [18,19,20,21,22,23]. In addition to mine activities, land use is also an important factor affecting concentration and distribution of heavy metals in the soil [24,25,26]. For example, the soil over the Xiaoqinling gold-mining region was polluted by Hg, Pb, Cu and As, among which Hg, Pb and Cu pollution was caused by gold-mining activities, whereas As pollution was caused by agricultural activities [18]. Studies are yet to address the relation between soil heavy metal pollution and complex mine activities under different types of land use.

The Xikuangshan (XKS) mine, located in the central Hunan province of China, is well known as the world’s largest antimony mine (Figure 1a). The long history of Sb-related mining and smelting activities caused high concentrations of Sb, As, Cd, Hg, Pb and Zn in soils near the mining area [27,28,29,30], which posed a great threat to the paddy soils along the Zijiang River basin (Figure 1b) [31]. However, the previous studies paid more attention to soil heavy metals pollution at the mining area, without considering land cover affecting spatial distribution of heavy metals around the mining area. Consequently, this study is aimed at systematically investigating the distribution of soil heavy metals in the wider mining-affected area from different land use patterns. The main specific objectives are (1) to determine soil heavy metal concentrations around the XKS mine area and evaluate their potential ecological risks, (2) to reveal the influence of land cover on distribution of heavy metal concentrations, and (3) to identify potential sources of heavy metals in soils by multivariate statistics combined with GIS spatial analysis and land use influence assessment.

## 2. Materials and Methods

### 2.1. Study Area

The XKS antimony mine area is located near the Lengshuijiang city of Hunan Province, south-central China. It is one of the world’s largest antimony mines with a super large deposit and more than one hundred years of exploration [32], as shown in Figure 1a. Stibnite (Sb_2_S_3_) is the primary ore mineral in the XKS mine, accompanied by trace amounts of pyrite (FeS_2_), arsenopyrite (FeAsS), sphalerite (ZnS) and galena (PbS) [33]. The XKS mine consists of a closed North Mine and an active South Mine, which processes local antimony ore and is involved in the smelting of ores from different locations of origin. Many Sb, Pb and Zn smelters are spreading over the XKS mine area. These activities related to mining, beneficiation and smelting have generated large amounts of solid wastes, including barren rock, fine-grained ore mineral, tailings and smelting slag. Different small plots such as farmland, grassland, woodland and bare land surround the mining area.

The XKS mine is located in the middle subtropical monsoon climatic zone, with average annual temperature, wind speed and precipitation of 16.8 ℃, 1.6 m/s and 1354 mm, respectively. The rainy season mainly occurs from March to August. The prevailing wind direction is primarily NW, with increased NNE and WWN components in the study area [34]. Several streams such as the Xuanshan, Qingfeng, Feishuiyan flow through the XKS mine and feed into the Zijiang River which is a first-class tributary of the Yangtze River (Figure 1b,c).

### 2.2. Soil Sampling and Measurement

A total of 38 sampling sites were distributed over the study area (Figure 1c), including 13, 13, 8 and 4 sites in farmland, grassland, woodland and bare land, respectively. The classification of land cover was determined by field survey of land vegetation when soil sampling. Approximately 1.0 kg of each sample was composed of four sub-samples of topsoil (5–15 cm depth) that were collected by a stainless steel shovel from four points within a 5 m radius of the sampling sites. The collected topsoil samples, when transported to the laboratory, were air-dried, ground, sieved to pass through 2 mm polyethylene mesh, and stored in a clean desiccator for the next treatment. Parts of these soil samples were further milled with a carnelian mortar and passed through a 0.15 mm plastic sieve for chemical analysis.

A portion of soil samples (0.5 g for each) were completely digested using the HCl-HNO_3_-HF-HClO_4_ method for determination of Cu, Zn, Cr, Cd, Pb and Ni concentrations, according to the Chinese Ministry of Environmental Protection (CMEP) method HJ 781-2016 [35]. The total concentrations of the above metals in the extracts were analyzed by inductively coupled plasma mass spectrometry (ICP-MS). Another portion of soil samples (about 0.2 g) were digested using aqua regia (HNO_3_:HCl = 1:3, v/v), following standard procedures of the CMEP method HJ 680-2013 [36]. After that, Hg, As and Sb concentrations were determined by atomic fluorescence spectrometry (AFS). Analytical data quality was verified using quality assurance and quality control that included the analysis of reagent blanks, duplicate samples and standard reference materials for each batch of samples. The recoveries of targeted heavy metals ranged from 85 to 115%. The error of the replicated samples analysis was within ±10%.

### 2.3. Statistical Analysis

Descriptive statistics were used to characterize variability and distribution of heavy metal concentrations in the soil. The differences in heavy metal concentrations were compared between the different land cover types using the one-way analysis of variance (ANOVA) [37,38]. Pearson correlation analysis was applied to identify the correlation among the heavy metals, which provided effective information to interpret their source relation [31,39,40]. The principal component analysis (PCA) was used to find two or three aggregated variables that controlled the heavy metal concentrations, and further assisted with identifying which natural or anthropogenic factors control heavy metal concentrations in soil [31,40,41]. The cluster relationship between heavy metals was visually demonstrated by the hierarchical cluster analysis (HCA) which could infer certain source association among heavy metals [14,28,31]. These multivariate statistical analysis approaches were performed using SPSS software. The Ordinary Kriging interpolation was calculated using the geography information system (GIS) software to simulate the spatial distribution of heavy metal concentrations and their potential ecological risk over the study area [38,40]. The predictive value error analysis (Appendix A) was used to evaluate the interpolation applicability.

### 2.4. Pollution and Risk Assessment Analysis

The Nemerow’s comprehensive index (*P_s_*) was employed to evaluate the soil comprehensive contamination status for all the heavy metals [28,31,42,43,44]. The *P_s_* calculated Formulas (1) and (2) are as follows:(1)$$PI_{i}=\frac{C_{s}^{i}}{C_{n}^{i}}$$
(2)$$P_{s}=\sqrt{\frac{{\left(PI_{i\;\mathrm{Avg}}\right)}^{2}+{\left(PI_{i\;\mathrm{Max}}\right)}^{2}}{2}}$$
where $C_{s}^{i}$ and *PI*_i_ represent the single pollution index of a given element i, where $C_{s}^{i}$, $C_{n}^{i}$ are the measured concentration and corresponding background reference value of the given element i, respectively. The *PI*_i Avg_ and *PI*_i Max_ are the average and the maximum value of *PI*_i_ for all the considered heavy metals, respectively. The degree of pollution according to the *P_s_* value was classified into five classes (Appendix A).

The potential ecological risk index (*RI*) was used to evaluate the all-round potential ecological risk due to heavy metal accumulation in soils [8,14,26,31,40,45]. The integrated *RI* is calculated by the following Formulas (3) and (4):(3)$$E_{r}^{i}=T_{r}^{i}\times \frac{C_{s}^{i}}{C_{n}^{i}}$$
(4)$$RI=\sum_{i=1}^{n}E_{r}^{i}$$ where $E_{r}^{i}$ represents the potential ecological risk index for a single element. $T_{r}^{i}$ is the toxic response factor (Cd = 30; As and Sb = 10; Cu, Ni, and Pb = 5; Cr and V = 2; Mn and Zn = 1). Appendix A summarized the classification of $E_{r}^{i}$ and *RI* grades for heavy metal contamination.

## 3. Results and Discussion

### 3.1. Pollution Degree and Potential Ecological Risk of Soil Heavy Metals

#### 3.1.1. Concentrations of Soil Heavy Metals

Basic statistical characteristics of heavy metal concentrations in topsoil samples from the XKS area were summarized in Table 1. The average concentrations of Sb, As, Cd, Cr, Hg, Pb, Cu, Zn and Ni in the soils were 633 mg/kg, 61 mg/kg, 3.2 mg/kg, 77.1 mg/kg, 1.7 mg/kg, 43.0 mg/kg, 39.6 mg/kg, 188.9 mg/kg and 38.8 mg/kg, respectively. The average concentrations of the targeted heavy metals were higher than their background levels of soils on the Hunan Province scale [46], as well as in China [47]. The Sb concentration at all sampling sites was higher than its background reference value, while concentrations of other heavy metals at 53% to 92% of sampling sites exceeded their background reference values. The coefficient of variation of Sb, Hg, Cd and Zn was relatively high (101.5%~265.8%), suggesting their distribution was variable in the soils of the study area. The concentrations of these heavy metals had positive kurtosis (9.9~24.2) and skewness values (2.9~4.7), illustrating that they were steeper and lower than the normal distribution.

According to the environmental quality standards for soils in China (GB 15618-2018) [48], heavy metal concentrations exceeding the corresponding risk screening values mean there is existing potential risk to the quality and safety of agricultural products, crop growth or soil ecological environment. The concentrations of Sb, As and Cd higher than the threshold values accounted for 71%, 71% and 53% of the total collected samples, respectively. The maximum Sb, Cd and As concentrations were 178-fold, 124-fold and 6.4-fold as high as their risk screening values, respectively. Therefore, it is suggested that heavy metals in the soils of the study area might have external sources and might be enriched to different degrees with sites.

#### 3.1.2. Pollution Level and Potential Ecological Risk of Heavy Metals

In order to give an assessment of the overall pollution status for a sample, the Nemerow’s integrated pollution index (*P_s_*) can be employed. As illustrated in Figure 2a, Ps values ranged from 1.2 to 1276, suggesting all sampling sites were contaminated by the heavy metals. Overall, 86.8%, 7.9% and 5.3% of samples were severely polluted (Ps > 3.0), moderately polluted (2.0 < *P_s_* ≤ 3.0) and slightly polluted (1.0 < *P_s_* ≤ 2.0), respectively. In addition, the sampling sites near the waste rock and smelting slag heaps were subjected to extremely severe pollution of heavy metals, with Ps values higher than 100.

The *RI* integrates the ecological risk that takes the toxicology of different heavy metals into consideration, thereby providing a better evaluation of the potential risk [49]. As shown in Figure 2b, the *RI* values in all the sampling sites ranged from 52 to 20545, with the largest contributor from *E_r_* values of Sb. The top three of the highest mean *E_r_* values were Sb (2125), Cd (1123) and Hg (768), significantly higher than the other heavy metals. Overall, 57.9%, 44.7%, 36.8% and 2.6% of the total samples had very high potential ecological risk (*E_r_* ≥ 320) for Sb, Cd, Hg and Pb, respectively. All samples had low potential ecological risk for Cr, Zn, Ni and Cu, with Er values lower than 40. The proportion of samples with low risk from As, Hg, Cd, Pb and Sb were 71%, 47%, 45%, 34% and 11%, respectively.

According to RI values classification, 68.4%, 5.3%, 7.9% and 18.4% of samples (Figure 2b) were very high, considerable, moderate and low potential ecological risks, respectively. The very high ecological risk area was principally distributed in the center of the north–south strip (Appendix A), and overlapped with the goaf area of the antimony mine where a large number of waste rock and smelting slag heaps were distributed. The low ecological risk area was distributed in the eastern and western edges away from the mining impacted area. It is indicated that the high RI values in the study area were closely related to mine activities. Therefore, it is essential to recognize hot-spot areas with high concentrations of heavy metals and associated sources.

### 3.2. Spatial Distribution of Heavy Metal Concentrations

The spatial distribution of heavy metals in soils is associated with the natural weathering of parent materials or anthropogenic activities [50]. The Ordinary Kriging interpolation method was performed to visualize the spatial distribution of the targeted heavy metals in the soils of the study area (Figure 3). The overall tendency of Sb concentrations in soils was reduced from northeast to southwest. Three hot-spots with extensively high concentrations of Sb, higher than 2000 mg/kg, were clearly observed in the vicinity of large waste rock and smelting slag heaps in the mined-out area. The higher concentrations of As were distributed over the impacted area by waste rock and smelting slag heaps; whereas, lower concentrations of As were distributed in the northern, eastern and southwestern edges. The leaching of arsenic alkali residue from stibnite smelters was considered as a crucial source of As in soils [30,51].

The concentrations of Pb in the southeast were higher than those in the north, with a hot-spot located in the central area near a smelting slag heap. This is consistent with the distribution of galena smelters in the South mine. The reported general sources of Pb were vehicle exhaust and industrial fumes [52], which can explain why higher concentrations of Pb were distributed in the main road sides along the Zhonglian river. The spatial variation of Hg was gradually reduced from the middle to the surrounding area, with a relatively large centralized region of higher Hg concentration in the center. Hg was commonly believed to come from coal combustion and atmospheric deposition [53]. The hot-spot area with a high concentration of Hg was mostly located downwind of the smelters in the North mine.

Cd and Zn hotspots were located, respectively, at the Tan’s creek in the North mine and the Feishuiyan stream in the South mine. The enrichment of Cd and Zn in topsoil might be related to the wide application of Cd and Zn in agricultural production [54,55,56]. In addition, the hot-spot with the highest concentration of Zn (1206 mg/kg) was distributed around the sphalerite smelting slag heaps in the South mine. Spatial distribution trends for Cu, Cr and Ni in soils were highly similar, approximately decreasing from the southwest to the northeast. There were no obvious anthropogenic sources, although Cu, Cr and Ni had higher concentrations around the tailings pond. Additionally, Cu, Cr and Ni had relatively lower coefficients of variation and average concentrations close to their background contents. Therefore, their primary source was indicated to be from natural parent materials.

### 3.3. Influences of Different Land Cover Patterns on Heavy Metal Concentrations

Land cover is an important factor affecting the distribution and accumulation of heavy metals in soils [24,26,57]. According to surrounding land vegetation, the land cover around the mining area where soil samples were collected included plowland, grassland, woodland and bare land. One-way analysis of variance was employed to confirm whether land cover had any effect on the concentrations of heavy metals in the soils of the study area. Tukey test and Tamhane’s T2 test were applied to Sb, Cu and Zn with homogeneity of variance and other heavy metals (As, Hg, Cr, Cd, Pb and Ni) with heterogeneity of variance, respectively. The analysis results are shown in Table 2. Concentrations of Hg, Pb and Cd are different among land cover patterns, with relatively higher F values and *p* < 0.05. The concentrations of Hg and Pb were significantly higher in the soils of bare land than in those in grassland and woodland, reflecting that wind-borne transport may play a role in their spatial spreading. The forest grassland was observed to be less influenced by human inputs and its heavy metal concentration was relatively low [25]. Cd concentration in the plowland soil was significantly higher than that in other types of land, suggesting that Cd could be caused by agricultural practices. The land cover exerted no significant influence on the concentrations of Sb, As, Cr, Cu, Zn and Ni, with relatively lower F values and *p* > 0.05. Combined with their spatial variation, it can be inferred that Cr, Cu and Ni in soil were primarily controlled by parent materials.

### 3.4. Potential Sources of Heavy Metals in Soils

Pearson’s correlation coefficient analysis, PCA and HCA were used to identify the potential sources of the targeted heavy metals in the soils of the study area. The results of PCA, shown in Table 3, suggested that three principal components with eigenvalues of 1.2~2.8 explained 71.6% of the total variance. Appendix A summarized the values of Pearson’s correlation coefficient matrix. The hierarchical dendrogram (Figure 4) showed that four clusters could be identified, with cluster I including Hg, Pb and As, cluster II including Sb, cluster III including Cd and Zn, and cluster IV including Cr, Ni and Cu.

As shown in Table 3, the first principal component (PC1) included Pb, As, Hg and Zn that had high factor loading (≥0.648), and accounted for 31.6% of the cumulative variance. The PC1 which mainly controlled concentrations of Pb, As, Hg and Zn was probably associated with the smelting process of sulfide minerals such as stibnite, galena and sphalerite in the XKS area. Pb, As and Hg are significantly correlated (*p* < 0.01) with each other (r = 0.822, 0.505, 0.403 for Pb and Hg, Pb and As, As and Hg, respectively), and are classified into the cluster I. Zn is moderately correlated (*p* < 0.05) with Pb (r = 0.329) and As (r = 0.391). The strong correlations among these heavy metals potentially indicate that they are of similar origin, which is further evidence that the smelting process of sulfide minerals is their primary source.

The second principal component (PC2) contributed to 25.6% of the cumulative variance, and mainly included Ni, Cr and Cu that had high factor loadings of 0.855, 0.836 and 0.567, respectively. The PC2 indicated a natural factor. Cr was significantly (*p* < 0.01) correlated with Ni (r = 0.768) and moderately correlated (*p* < 0.05) with Cu (r = 0.630), which was in accordance with the results of cluster IV. A relatively uniform trend of spatial distribution of Ni, Cr, and Cu concentrations was shown in Figure 3. Additionally, the concentrations of Ni, Cr, and Cu in the study area were close to their corresponding background reference values with relatively lower CV, skewness and kurtosis (Table 1). Therefore, Ni, Cr and Cu could be mainly obtained from natural weathering of geological parent materials.

The third principal component (PC3) accounted for 14.4% of the total variance, which had high factor loadings for Cd (0.748) and Zn (0.577). Cd was significantly (*p* < 0.01) correlated with Zn (r = 0.470), which was consistent with the results of cluster III. Therefore, Cd and Zn may be controlled by common sources. Zn is an effective ingredient of fertilizers and bactericides used to feed and cash crops [50,58]. Cd is a common element of phosphate fertilizer [59,60]. Animal manure is often considered to be another source of Zn and Cd in topsoil [61]. This is consistent with the difference in land cover patterns. The plowland soils had significantly higher Cd and Zn concentrations than woodland, grassland and bare land (Table 2). Therefore, agricultural practice was an important factor (PC3) in the accumulation of Cd and Zn in soil. However, sphalerite smelting was also another major source of Zn, as discussed in the PC1 and spatial distribution section. That is also why land cover had a less significant effect on Zn than Cd.

According to the HCA results (Figure 4), Sb was of an individual cluster (cluster II) and could belong to a higher cluster with Hg, Pb and As, suggesting that Sb was also derived from another source different from Hg, Pb and As. The PC1 had a lower factor loading of 0.480 for Sb than that for Hg, Pb, As and Zn. In addition, Sb was not significantly correlated with any heavy metals. Therefore, Sb concentrations in soils were a result of the comprehensive influence of multiple factors in the XKS area. Excluding smelting slags, quantities of soluble and insoluble Sb released into water and soil could be also derived from mine drainage, leaching of waste rock and runoff erosion of tailings [62,63,64,65,66]. Moreover, Sb can be released to the atmosphere attached to fine particulates during high temperature smelting processes. The transport and deposition of fine particulates are controlled by the prevailing wind. To reveal the influence of fume emission and aeolian erosion on Sb source, a clear decline in Sb concentration was observed with increasing distance from the Sb smelters [34]. High Sb concentrations were also observed downwind of the major Sb smelters in this study. Therefore, Sb concentrations in soils were derived from complex mine activities in the XKS area including mining, beneficiation and smelting. However, the apportionment of Sb sources associated with the mine activities needs to be further determined.

## 4. Conclusions

The contamination characteristics of heavy metals in soils around the XKS mine area were characterized by the combined application of statistical analysis, Nemerow’s comprehensive index and potential ecological risks. The average concentrations of Sb, As, Cd, Cr, Hg, Pb, Cu, Zn and Ni in soils of the XKS antimony mine area were higher than their corresponding background levels. A total of 71%, 71% and 53% of samples had Sb, As and Cd exceeding their risk screening values, respectively. Additionally, 86.8% of samples were severely polluted, characterized by the Nemerow’s comprehensive index, and 68.4% of samples were of very high potential ecological risk primarily contributed by Sb, Cd, Hg. The ANOVA results revealed that land cover had a significant influence on Hg, Pb and Cd accumulation in soils. Pearson correlation, PCA and HCA combined with spatial distribution of the targeted heavy metals in soils indicated that Ni, Cr and Cu were derived mainly from natural parent materials, while other heavy metals were derived mainly from anthropogenic sources. Pb, As and Hg were mainly derived from smelting processes of sulfide minerals in the XKS area. Cd and Zn mainly resulted from agricultural activities and Zn was also affected by sphalerite smelting. The spatial distribution of Sb concentrations in soils was controlled by multiple factors including mining, beneficiation and smelting activities and related processes. Therefore, the reasonable disposal of smelting waste slag and wastewater can significantly reduce Pb, As, Hg and Zn pollution. More attention should be paid to effective control strategies of soil Sb pollution caused by comprehensive antimony mining activities.

## Figures and Tables

**Figure 1 ijerph-20-02177-f001:**
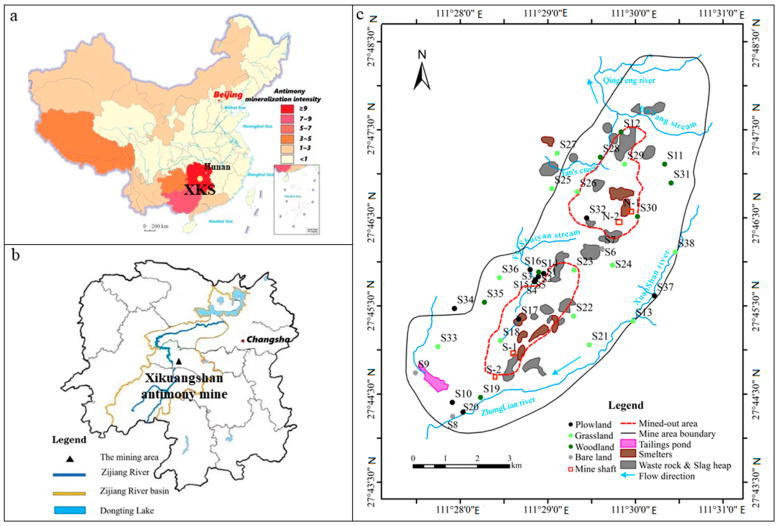
The location map of the study area and distribution of soil sampling sites. (**a**) antimony mineralization intensity in China and location of the Xikuangshan mine (XKS) in China; (**b**) the XKS antimony mine location in the Zijiang River basin and Hunan province; (**c**) distribution of mine activities in the XKS mine area and soil sampling sites of this study.

**Figure 2 ijerph-20-02177-f002:**
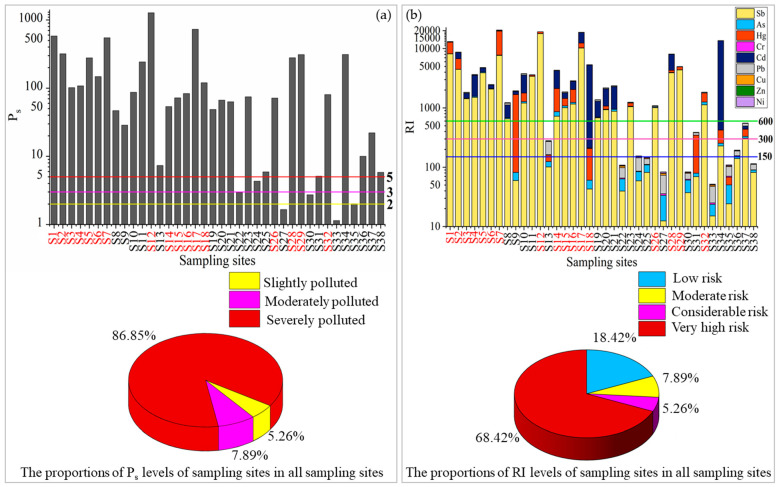
Values and classification proportion of the Nemerow’s comprehensive index (*Ps*) (**a**) and the potential ecological risk index (*RI*) (**b**) for heavy metals in soils of the study area. The sites which are located in the mining area are labeled by coloring the IDs in red.

**Figure 3 ijerph-20-02177-f003:**
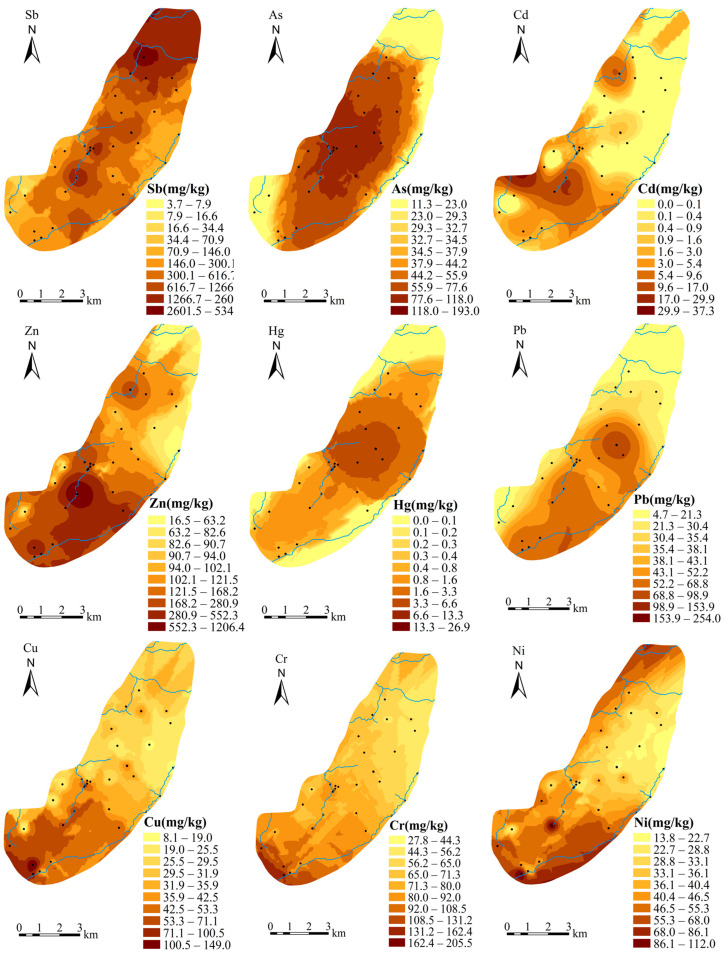
Distribution maps of heavy metals (Sb, As, Cd, Zn, Hg, Pb, Cr, Cu and Ni) in soil over the study area.

**Figure 4 ijerph-20-02177-f004:**
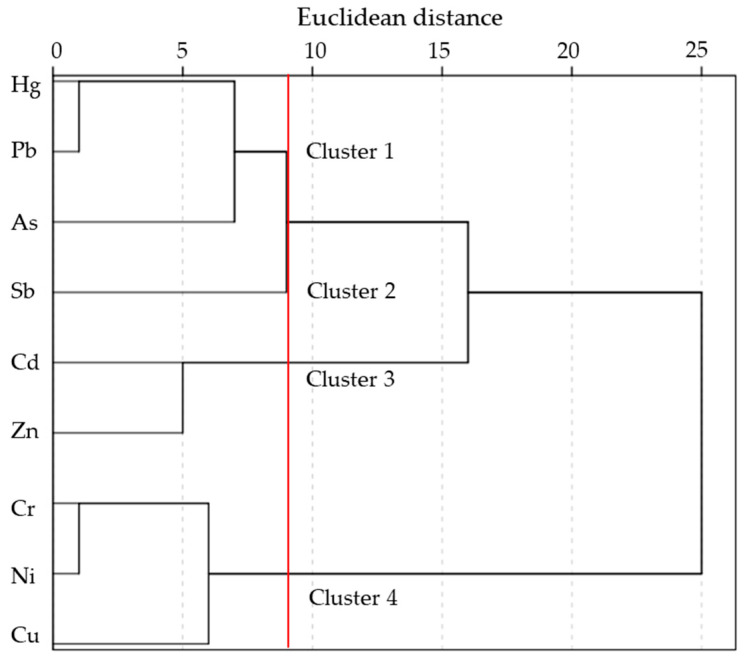
Dendrogram obtained by HCA (Ward’s method) for heavy metal concentrations in soils of the study area.

**Table 1 ijerph-20-02177-t001:** Statistical data for soil heavy metal concentration (mg/kg) of the study area (n = 38).

Heavy Metal	Minimum Values	Maximum Values	Mean Values	Median Values	Coefficient of Variation (%)	Skewness	Kurtosis
Sb	3.7	5343	633	269	167.2	2.9	9.9
As	11.3	193	61	35.9	84.9	1.5	1.1
Cd	n.a.	37.3	3.2	0.7	215.7	3.7	15.9
Cr	27.8	206	77.1	70.5	42.7	2.1	5.8
Hg	n.a.	26.9	1.7	0.26	265.8	4.7	24.2
Pb	4.7	254	43.0	34.8	94.9	3.9	18.6
Cu	8.1	149	39.6	32.6	62.7	2.4	8.8
Zn	16.5	1206	188.9	124.2	101.5	4.1	20.9
Ni	13.8	112	38.8	35.8	54.3	2.1	5.1
**Heavy Metal**	**Average Soil Background** **Values in Hunan Province ^1^**	**Average Soil Background** **Values in China ^2^**		**Risk Screening Values for Soil Contamination (RSV) ^3^**	**Percentage of Exceeding** **Hunan Background Values (%)**		**Percentage of Exceeding** **RSV (%)**
Sb	2.98	<1.0		30 ^4^	100		71.1
As	14	11.2		30	92.1		71.1
Cd	0.085	0.097		0.3	55.3		52.6
Cr	67	61		200	63.2		2.6
Hg	0.09	0.065		2.4	52.6		18.4
Pb	27	26		120	63.1		2.6
Cu	26	22.6		100	76.3		2.6
Zn	94	74.2		250	73.7		23.7
Ni	31.9	26.9		100	57.9		5.3

^1^ The average soil background values for heavy metals in Hunan Province of China [46]. ^2^ The average background values of heavy metals in soil in China [47]. ^3^ Chinese Environmental Protection Administration (GB 15618-2018) [48]. ^4^ The RSV of Sb is currently not available and is assumed to be the same as As in this study.

**Table 2 ijerph-20-02177-t002:** Results of ANOVA for soil samples among different land cover patterns.

Element	Statistic	Plowland	Grassland	Woodland	Bare Land	*F* Value	Sig. (*p*-Value)
Sb ^1^	mean	817.26	180.43	997.45	778.33	1.257	0.305
	CV (%)	937.61	375.16	1814.89	1033.37		
As ^2^	mean	84.85	34.59	58.61	71.13	2.284	0.097
	CV (%)	56.99	22.00	59.65	68.85		
Hg ^2^	mean	2.00	0.15	0.91	7.61	3.222	0.035
	CV (%)	3.03	0.34	1.20	12.94		
Cr ^2^	mean	77.22	66.89	68.47	87.50	1.975	0.141
	CV (%)	25.05	24.37	9.57	28.02		
Cd ^2^	mean	6.20	1.39	2.28	1.00	2.912	0.046
	CV (%)	10.42	4.03	3.93	0.32		
Pb ^2^	mean	52.65	29.98	26.79	86.76	2.972	0.045
	CV (%)	25.24	15.97	14.99	112.53		
Cu ^1^	mean	51.70	30.14	26.16	58.16	1.796	0.119
	CV (%)	33.14	14.10	9.21	21.78		
Zn ^1^	mean	276.46	125.01	171.85	146.13	1.492	0.234
	CV (%)	292.22	81.65	125.63	78.24		
Ni ^2^	mean	29.81	43.67	36.66	56.25	2.081	0.121
	CV (%)	7.23	15.27	25.40	44.41		

^1^ Tukey test. ^2^ Tamhane’s test.

**Table 3 ijerph-20-02177-t003:** Rotated component matrix of heavy metals by PCA.

Element	Components		
	1	2	3
Sb	0.480	−0.437	0.079
As	0.684	−0.195	0.000
Hg	0.665	−0.476	−0.423
Cr	0.308	0.836	−0.341
Cd	0.343	0.137	0.748
Pb	0.834	−0.218	−0.286
Cu	0.607	0.567	−0.049
Zn	0.648	0.164	0.577
Ni	0.102	0.855	−0.132
Eigenvalues	2.847	2.300	1.295
% of variance	31.638	25.552	14.387
% of cumulative	31.638	57.190	71.577

## Data Availability

The data that support the findings of this study are available from the corresponding author, Xiaoqian Li (lixiaoqian@cug.edu.cn), upon reasonable request.

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
