# Peer review of "Heavy Metals in Soil around a Typical Antimony Mine Area of China: Pollution Characteristics, Land Cover Influence and Source Identification"

_ijerph, 2023, doi:10.3390/ijerph20032177_

Round 1
Reviewer 1 Report
Dear authors,
Firstly I didn't find the aim in abstract chapter.
I suggest to clearify the metodology chapter, because it not clear which tools were used to take soil samples and which equipment was used to sieve soils samples.
I misses a reference in statistical analyses chapter, because you couldn't invent statistical methods which you described.
I think your conclusion could be better if you involve some suggestions how to reduce pollution and what measures to use.
You have to check how to submit references.
Author Response
- Firstly I didn't find the aim in abstract chapter.
Response:The first sentence in the abstract (“To understand contamination characteristics and identify sources of heavy metals in soil affected by complex mine activities”) is the aim of this study.
- I suggest to clearify the metodology chapter, because it not clearwhich tools were used to take soil samples and which equipment was used to sieve soils samples.
Response:The details has been added to clearify the sampling and sieve tools in line 100-102. (soil samples were collected by a stainless steel shovel and samples were sieved to pass through 2 mm polyethylene mesh )
- I misses a reference in statistical analyses chapter, because you couldn't invent statistical methods which you described.
Response:have added references in the statistical analyses chapter (line 132-144).
- I think your conclusion could be betterif you involve some suggestions how to reduce pollution and what measures to use.
Response:have added suggestions in the conclusion (line 357-359). “Therefore, reasonable disposal of smelting waste slag and wastewater can significantly reduce Pb, As, Hg and Zn pollution. More attention should be paid to effective control strategies of soil Sb pollution caused by comprehensive antimony mine activities.”
- You have to check how to submit references.
Response:revised references format according to the requirements of IJERPH.
Reviewer 2 Report
The paper aimed to investigate soil contamination and the factors surrounding the mine area in China.
Overall, the background information is not sufficiently provided. Therefore, it is hard to determine if the discussed associations are reasonable. Here are the comments to be addressed.
1. Figures resolutions are low. They need to improve the quality.
2. Figures must show only the relevant information. For such as Figure 1(a), what information is intended to deliver? The figure should contain only that information and should be explained in the main body.
3. Equation (1) must miss some information. (csi/cni)max needs the information on i.
4. Equation (2) must miss some information. r is for what?
5. Ps and RI need more literature review, particularly about the associations between these indexes and actual negative impacts on the environment and human health in other cases.
6. What is the source of land use information? And how was the correlation calculated? In any case, the sample size seems too small to conclude any deterministic associations.
7. I don't understand why the correlation analysis, PCA, and HCA were all applied. After all, the authors' intention seems to want to know the associations. If so, the correlation analysis must suffice.
8. L259-260: selection of the number of clusters is arbitrary. Same as to the above comment, the authors need to elaborate on the intention of each analysis method.
Author Response
The paper aimed to investigate soil contamination and the factors surrounding the mine area in China. Overall, the background information is not sufficiently provided. Therefore, it is hard to determine if the discussed associations are reasonable.
Response:The paper is aimed to address the relation between soil heavy metal pollution and complex mine activities under different types of land use around a typical antimony mime. I revised the introduction (line 51-58) to make it clear and reasonable.
Here are the comments to be addressed.
- Figures resolutions are low. They need to improve the quality. 图件分辨率
Response:The low figures resolution might be caused when inserting them into the word document. I replaced all figures with high quality ones.
- Figures must show only the relevant information. For such as Figure 1(a), what information is intended to deliver? The figure should contain only that information and should be explained in the main body.
Response:Figure 1 has been simplified to contain clear information. Figure 1a is aimed to show the typicality of the XKS antimony mine in China; Figure 1b shows the location of the XKS mine in the regional Zijiang River basin.
- Equation (1) must miss some information. (csi/cni)max needs the information on i.
Response:(csi/cni)max is the maximum value among pollution indexes of all the considered metals, which is added to explain it clear (line 152-153).
- Equation (2) must miss some information. r is for what?
Response:Here Er and Tr are whole signs to represent potential ecological risk index toxic response factor, respectively, which are cited from previous studies (references [8,14,26,31,40, 45] ).
- Ps and RI need more literature review, particularly about the associations between these indexes and actual negative impacts on the environment and human health in other cases.
Response:The other cases applying Ps and RI to describe the heavy metal pollution and adverse effect has been added in the references.
- What is the source of land use information? And how was the correlation calculated? In any case, the sample size seems too small to conclude any deterministic associations.
Response:The land use was identified by field investigation when collecting soil samples according to the local landscape such as plowland, grassland, woodland and bared land. The correlation was calculated by the one-way analysis of variance using SPSS software. The sample size is generally more than 10. For small sample size, the influence of outliers is particularly significant. Therefore, the analysis results of this study can reflect whether the land use had significant influence.
- I don't understand why the correlation analysis, PCA, and HCA were all applied. After all, the authors' intention seems to want to know the associations. If so, the correlation analysis must suffice.
- L259-260: selection of the number of clusters is arbitrary. Same as to the above comment, the authors need to elaborate on the intention of each analysis method.
Response to comment 7 and 8 together: The combined application of correlation analysis, PCA and HCA was to identify potential sources of soil heavy metals. The intention of each analysis method had been elaborated in line 128-140. Each method can provide information to a common source that is associated with different heavy metals, but the interpretation of the result is subjective to a certain extent. However, the mutual corroboration of the three methods makes the source analysis more adequate, which have proved to be a powerful tool for identifying sources of heavy metal contamination by previous studies.
Round 2
Reviewer 1 Report
I appreciate the authors' effort in adopting the corrections that were suggested.
Author Response
I appreciate greatly for the reviewer‘s approval of the last revised manuscript.
Reviewer 2 Report
I found the paper improved, but the following points are yet to be addressed.
1. Figures resolutions are again low. Now I barely see the figure contents.
2. >(csi/cni)max is the maximum value among pollution indexes of all the considered metals
If so, it should be maximized in terms of i. It should be written in a mathematically correct way.
3. Ps and RI reflated literature were added, which I appreciate; however, not only listing the articles but preferably introducing several application examples and implications may be added, which is helpful for readers.
4. I think "land cover" better fits than "land use" considering the used classification. If the authors determined the classes without authorized reference information, determination criteria might need to be elaborated.
5. ANOVA can test if there are differences among the levels but not which differences are significant.
6. As the authors described that PCA is for determining aggregated variables, but the author did not well utilize these aggregated variables. Such as the intention of "The strong correlations among these heavy metals potentially indicate their similar origins, demonstrating that an anthropogenic source for PC1 was associated with the smelting processes of sulfide minerals such as stibnite, galena and sphalerite in the XKS area." is not clear.
7. Table 1: Percent of ... should be Percentage of samples exceeding...
8. Figure 2: It is helpful if the sites which are close to the mine areas can be identified, such as by coloring the site IDs.
Author Response
Response to Reviewer 2
- Figures resolutions are again low. Now I barely see the figure contents.
Response: The font size in the figures has been increased according to the appropriate layout to improve the figures resolution. Fig.2, Fig.3 and Fig.4 has been revised.
- >(csi/cni)max is the maximum value among pollution indexes of all the considered metals. If so, it should be maximized in terms of i. It should be written in a mathematically correct way.
Response: The expression of (csi/cni)max was directly cited from the published references, which might be written in a mathematically wrong way. in order to facilitate the understanding, We changed the Ps formula into another way defined as [0.5*(PI i Avg)2+(PIi Max)2]1/2, where the mathematical processes for calculating the average and the maximum values of PIi are omitted.
- Ps and RI reflated literature were added, which I appreciate; however, not only listing the articles but preferably introducing several application examples and implications may be added, which is helpful for readers.
Response: I am sorry we still haven’t added several application examples of Ps and RI. The reasons are as follows: (1) the definition and meaning of Ps and RI has been elaborated in detail in the section 2.4,which can already help readers to understand them clearly; (2) the classification of Ps grades as well as and RI for heavy metal contamination has been summarized in Table S2 and Table S3, which is the evaluation basis for application cases; (3) Ps and RI has been widely used to quantitatively evaluate pollution levels and potential ecological risk of soil heavy metals;(4) the application case in this study is no difficult to understand without introduction of another several examples.
- I think "land cover" better fits than "land use" considering the used classification. If the authors determined the classes without authorized reference information, determination criteria might need to be elaborated.
Response: I totally agree with the above suggestion “land cover better fits than land use ” in this study. The “land use” has been replaced with “land cover” in the revised ms. “The classification on land cover was determined by field survey of land vegetation when soil sampling” has been added in lines 98-99.
- ANOVA can test if there are differences among the levels but not which differences are significant.
Response: I think ANOVA can test if there are differences among the levels but also which differences are significant. As like ANOVA results in Table 2, F values can evaluate the difference among the levels, the higher F value is, the bigger the difference is. And p value (p<0.05) can test if the difference is significant among the levels. Therefore, the ANOVA method can test if there is significant difference among the levels, thus to determine if the land cover has significant influence on the heavy metal concentrations. Therefore, revision is as an example of “Concentrations of Hg, Pb and Cd are different among land cover patterns, with relatively higher F values and p<0.05. ” (lines 281-282).
- As the authors described that PCA is for determining aggregated variables, butthe author did not well utilize these aggregated variables. Such as the intention of "The strong correlations among these heavy metals potentially indicate their similar origins, demonstrating that an anthropogenic source for PC1 was associated with the smelting processes of sulfide minerals such as stibnite, galena and sphalerite in the XKS area." is not clear.
Response: Revised to make the intention of PCA results clear. The three principal components with eigenvalues of 1.2~2.8 explained 71.6% of the total variance are determined by PCA, which are the aggregated variables. The first principal component (PC1) included Pb, As, Hg, and Zn that had high factor loading (≥0.648), and accounted for 31.6% of the cumulative variance. The second principal component (PC2) contributed for 25.6% of the cumulative variance, and mainly included Ni, Cr, and Cu that had high factor loading of 0.855, 0.836 and 0.567, respectively.The third principal component (PC3) accounted for 14.4% of the total variance, which has high factor loadings for Cd (0.748) and Zn (0.577). Therefore, the PC1, PC2 and PC3 indicated smelting processes of sulfide minerals, soil parent rock weathering, and agricultural practice are main factors controlling heavy metal concentrations in soil. These clear intention has been revised in lines 305-307, 315 and 333. However, The purpose of PCA in this study is to help identify sources of heavy metals in soil rather than interpreting the three aggregated variables. Therefore, the results of PCA, correlation analysis and HCA are discussed together to explain the potential sources one by one.
- Table 1: Percent of ... should be Percentage of samples exceeding...
Response: Revised.
- Figure 2: It is helpful if the sites which are close to the mine areas can be identified, such as by coloring the site IDs.
Response: Revised. The sites which are within the mining area are identified by coloring the site IDs in red in Figure 2.
